# Reinforced Nafion Membrane with Ultrathin MWCNTs/Ceria Layers for Durable Proton-Exchange Membrane Fuel Cells

**DOI:** 10.3390/membranes12111073

**Published:** 2022-10-29

**Authors:** Dongsu Kim, Yeonghwan Jang, Eunho Choi, Ji Eon Chae, Segeun Jang

**Affiliations:** 1School of Mechanical Engineering, Kookmin University, Seoul 02707, Korea; 2Department of Mobility Power Research, Korea Institute of Machinery & Materials, 156 Gajeongbuk-ro, Yuseong-gu, Daejeon 34103, Korea

**Keywords:** proton-exchange membrane fuel cell, Nafion, high performance, durability, reinforced membrane, ultrathin, multiwalled carbon nanotube, CeO_2_

## Abstract

For further commercializing proton-exchange membrane fuel cells, it is crucial to attain long-term durability while achieving high performance. In this study, a strategy for modifying commercial Nafion membranes by introducing ultrathin multiwalled carbon nanotubes (MWCNTs)/CeO_2_ layers on both sides of the membrane was developed to construct a mechanically and chemically reinforced membrane electrode assembly. The dispersion properties of the MWCNTs were greatly improved through chemical modification with acid treatment, and the mixed solution of MWCNTs/CeO_2_ was uniformly prepared through a high-energy ball-milling process. By employing a spray-coating technique, the ultrathin MWCNTs/CeO_2_ layers were introduced onto the membrane surfaces without any agglomeration problem because the solvent rapidly evaporated during the layer-by-layer stacking process. These ultrathin and highly dispersed MWCNTs/CeO_2_ layers effectively reinforced the mechanical properties and chemical durability of the membrane while minimizing the performance drop despite their non-ion-conducting properties. The characteristics of the MWCNTs/CeO_2_ layers and the reinforced Nafion membrane were investigated using various in situ and ex situ measurement techniques; in addition, electrochemical measurements for fuel cells were conducted.

## 1. Introduction

As hydrogen energy has been in the spotlight as the next-generation eco-friendly energy source for achieving carbon neutrality, proton-exchange membrane fuel cells (PEMFCs) have received great attention in recent years due to their eco-friendly operation, low operating temperatures, fast dynamic response, and compact size. Furthermore, it is possible to produce electricity continuously if the hydrogen supply is sufficient, contrary to other fluctuating renewable energy sources such as wind and solar [1,2,3,4]. However, their high fabrication cost and poor durability—which is caused by mechanical and chemical degradation of the membrane electrode assembly (MEA) during long-term fuel cell operation—hinder their commercialization in the market, making PEMFCs not economically viable [5]. Accordingly, the U.S. Department of Energy has set 2025 targets that require almost two times higher durability of up to 8000 h and a reduced MEA cost of $30/kW compared to the values currently achieved of 4130 h and $52/kW [6].

The mechanical degradation of MEAs is mainly caused by repeated wet/dry cycles. Under this operating condition, membranes suffer from significant volume expansion/shrinkage, but the electrodes attached to both sides of a membrane tend to relatively maintain their original dimensions [7,8,9]. Therefore, initially, well-constructed interfaces between the membrane and electrodes become severely damaged as the wet/dry cycles continue, resulting in edge failure at the interfaces and forming and propagating cracks [10]. For the construction of mechanically reinforced membranes with high dimensional stability, there have been many studies that employ porous expanded polytetrafluoroethylene (e-PTFE) sheets [11,12], electrospun polymer web [13], and organic/inorganic fillers such as SiO_2_ [14], TiO_2_ [15], MXene [16], graphene oxide [17], and multiwalled carbon nanotubes (MWCNTs) [18]. Among them, the method of impregnating ionomer into the e-PTFE has been widely adopted, and commercial products such as GORE-SELECT and Nafion XL membranes have been used as successful cases. However, the interfacial compatibility issue between hydrophobic e-PTFE fibers and hydrophilic ionomers still exists during repeated wet/dry cycles [19]. To overcome this issue, surface modification of the e-PTFE has been widely studied with plasma irradiation, polydopamine, and sodium naphthalene treatment to reduce the surface energy of the e-PTFE [20]. In the case of the MWCNTs, which are composed of concentrically rolled graphene sheets with a diameter in the nanometer range with a high aspect ratio and excellent mechanical properties (e.g., high stiffness), they have proved their efficacy in increasing Young’s modulus and dimensional stability of the membrane when incorporated inside the membrane [21,22]. However, during the casting and drying of the mixed solution of MWCNTs/electrolyte ionomer for membrane fabrication, agglomeration and entanglement issues were raised due to bulk solvent evaporation, and this severely reduced proton conductivity [23]. Furthermore, when the MWCNTs content exceeds the electrical percolation threshold, it causes an electrical short circuit (electric crossover) across the membrane [24,25]. Therefore, it is desirable to develop a strategy that takes advantage of MWCNTs while addressing these problems.

For chemical degradation, free radicals such as hydroxyl (HO^•^) and hydroperoxyl (HOO^•^) are generated from the reaction between hydrogen peroxide (H_2_O_2_), which is formed by gas crossover and insufficient oxygen reduction reaction, and the cations (e.g., Fe^2+^) released from bipolar plates or other system components such as humidifiers [26,27]. These radicals decompose main and side chains of the membrane, which results in membrane thinning and pinhole formation [28,29,30]. To alleviate chemical degradation from the radical attack, a strategy to incorporate CeO_2_ and MnO_2_ nanoparticles into a membrane has mainly been studied, and commercial films (e.g., Nafion XL) have been developed [31]. CeO_2_, generally known to have the best radical removal efficacy, has been widely used, and the radical scavenging mechanism is based on the regeneration properties in acidic media through the oxidation–reduction reaction between Ce^3+^ and Ce^4+^ ionic states and expressed as the following equations [32,33]:(1)$${\mathrm{Ce}}^{3+}+{\mathrm{HO}}^{•}+H^{+}\to {\mathrm{Ce}}^{4+}+H_{2}$$
(2)$${\mathrm{Ce}}^{4+}+H_{2}O_{2}\to {\mathrm{Ce}}^{3+}+{\mathrm{HOO}}^{•}+H^{+}$$
(3)$${\mathrm{Ce}}^{4+}+{\mathrm{HOO}}^{•}\to {\mathrm{Ce}}^{3+}+O_{2}+H^{+}$$

Although the composite membrane with radical scavengers (e.g., CeO_2_) exhibited excellent chemical durability and made the MEA long-term operation possible, agglomeration of nanosized CeO_2_ and substitution of a proton with Ce^3+^/^4+^ ion at the sulfonic acid group severely decreased the proton conductivity [34,35] as well as the migration of Ce^3+^/^4+^ ion into the electrode [28], resulting in performance loss [36]. To address the migration issue, an 18-crown-6-ether/cerium ions coordination structure (CRE/Ce) was developed [37]. While CRE, which is a unique substance that forms a coordination complex with cerium ions, exhibited Ce^3+^ capturing ability in the modified membrane mixed with electrolyte ionomer and CRE/Ce, it negatively affects the proton conductivity and water uptake capacity. This resulted in a severe initial performance drop. For alleviating the severe proton conductivity loss derived from incorporating the CeO_2_ all over the membrane, the incorporation of ultrathin CeO_2_ layers into the Nafion membrane surface has been recently reported based on the spin coating of CeO_2_ solution onto a transfer film and successive thermal lamination of the film into the membrane [38]. At the interface between the membrane and electrode, the ultrathin CeO_2_ layers can effectively alleviate the radical attack while minimizing performance loss. Moreover, instead of using a radical scavenger, a gas barrier layer was introduced at the outermost sides of the Nafion membrane through direct spin coating of 2D hexagonal boron nitride (hBN) on the membrane [39]. The ultrathin hBN showed an effective role in preventing gas crossover while securing proton conductivity. The reduction gas crossover enabled the alleviation of H_2_O_2_ generation; therefore, free radical generation was severely reduced. Although these recent approaches effectively minimized the decrease in proton conductivity and improved the chemical durability through interface engineering between the membrane and electrode where the radical attack to the membrane is initiated, it is still necessary to develop scalable and reliable techniques for making ultrathin functional layers for the membrane surface because the spin-coating method has difficulty in large-area and continuous membrane fabrication.

In this study, to construct mechanically and chemically robust MEA for the long-term operation of PEMFCs, a method to fabricate a highly durable membrane with ultrathin functional layers (∼450 nm) on both sides of the membrane through an optimized dispersion process and a scalable spray-coating process is proposed. By simply modifying a commercially available Nafion membrane, it was possible to compare the performance of the modified Nafion membrane with that of a reference Nafion membrane without concern of the repeatability and reliability issues for the membrane fabrication process. Through a chemical acid treatment process, the dispersion problem of MWCNTs with a high aspect ratio and hydrophobicity was resolved. In addition, the high-energy ball-milling process enabled a well-dispersed MWCNTs/CeO_2_ mixed solution. Furthermore, the optimized layer-by-layer stacking and spray-coating process provided a uniformly distributed MWCNTs/CeO_2_ layer onto the Nafion membrane surface while controlling thickness of the functional layers in the submicrometer scale. The modified Nafion membrane exhibited both an increase in mechanical properties and chemical durability and a decrease in gas crossover by acting as a gas barrier. Finally, the MEA with the reinforced membrane showed much higher durability and comparable initial performance because of the ultrathin and uniformly dispersed characteristics of the functional layers.

## 2. Materials and Methods

### 2.1. Preparation of Surface-Modified MWCNTs and the Mixed Solution of MWCNTs/CeO_2_

For better dispersion of the MWCNTs, chemical acid treatment was employed to create oxygen functional groups on the MWCNTs. The procedure was as follows: (1) Sulfuric acid (H_2_SO_4_) and nitric acid (HNO_3_) solutions were mixed at 90 and 30 mL, respectively, and 1 g of pristine MWCNTs (50–90 nm in diameter; Sigma-Aldrich, St. Louis, MO, USA) was added. (2) Stirring was applied for 12 h on a hotplate at 80 °C [22]. During this process, to prevent concentration change, a reflux cooler system was used. (3) After that, the acid-treated MWCNTs were thoroughly washed with ethanol and deionized (DI) water and filtered by using vacuum filtration with a porous Teflon membrane. (4) The filtered MWCNTs were separated from the Teflon membrane and the washing process was further conducted to entirely remove the residual acids. Next, to prepare two kinds of mixed solutions of MWCNTs/CeO_2_, the prepared MWCNTs, a commercial CeO_2_ solution (∼25 nm in diameter, 10 wt% CeO_2_ dispersed in water; Sigma-Aldrich), and DI water were mixed with weight ratios of MWCNTs and CeO_2_ of 4:1 and 2:1. The mixed solutions were transferred to Nalgene bottles which contained zirconia balls (diameter: 5 mm), and the high-energy mechanical ball-milling process was conducted at 300 rpm for 12 h to let the CeO_2_ nanoparticles physically adhere to the modified MWCNTs surface with the oxygen functional groups. After that, the mixed solution was put in a vial, and DI water and a Nafion ionomer (5 wt%; Sigma-Aldrich) were further added. The ionomer ratio to the solid (MWCNTs and CeO_2_) was set at 30 wt% to secure the proton transport network. Next, for better ionomer distribution while minimizing detachment of CeO_2_ from the MWCNTs, an ultrasonication process was conducted in under 1 min.

### 2.2. Fabrication of the MWCNTs/CeO_2_-Coated Membranes and the MEAs

For constructing the ultrathin MWCNTs/CeO_2_ layers on the Nafion membrane, the prepared mixed solutions were sprayed onto the Nafion (NR211, 25 µm; Dupont, Wilmington, DE, USA) membrane using the lab-made spray-coating system. The system was manufactured by combining an airbrush (nozzle size of 0.7 mm, GP-70) with a commercial three-dimensional (3D) printing machine (Ender-3, Creality 3D, Shenzhen, China). The membrane was placed on a modified hotplate fitted with a vacuum pump. The temperature was set at 70 °C and vacuum pressure was applied to prevent swelling of the Nafion membrane during the solution deposition process by quickly removing the residual solvent. For uniform deposition of MWCNTs/CeO_2_, the spray travel path was designed with four serpentine-like patterns of 36 × 36 mm (width by height), and between each serpentine-like pattern the spray travel path was rotated 90° and the center was shifted 0.5 mm. Moreover, the nozzle-to-substrate distance was set at 8.4 mm while keeping spraying pressure at 2 bar. For comparison, the reference membrane was an unmodified pristine Nafion membrane. To construct the MEA, a catalyst slurry was prepared by mixing a Pt/C catalyst (40 wt%; Johnson Matthey, London, UK), a Nafion ionomer (density of 0.874 g·mL^−1^, 5 wt%; Sigma-Aldrich), DI water, and isopropanol (IPA) (99.5%; Samchun, Pyeongtaek-si, Korea). The Nafion ionomer content was set at 23 wt% relative to the total solid content of the catalyst slurry. In the catalyst layer, the prepared slurry was further dispersed by ultrasonication for 30 min and sprayed onto the modified and pristine Nafion membranes by using the same spray-coating system and vacuum hotplate. The active area was set at 5 cm^2^ using a stainless-steel mask with a square hole of 2.23 × 2.24 cm (width by height), and the catalyst loading amounts for all the MEA were kept at 0.2 mg_Pt_·cm^−2^ for both the cathode and anode sides.

### 2.3. Fenton’s Test and Stress–Strain Measurements for the Prepared Membranes

Fenton’s tests were done according to the following steps: (1) The specimens were cut into 2 × 2 cm, and the weight of each specimen was measured after drying the specimens for 6 h in an oven at 80 °C. (2) The prepared specimens were immersed in a 60 mL Fenton solution (30 ppm Fe^2+^ in 20 wt% aqueous H_2_O_2_) for 48 h at 80 °C. (3) The fluoride ion concentrations eluted from the test specimens were determined using a calibrated fluoride ion-selective electrode (A214, Thermo Fisher Scientific, Waltham, MA, USA) by mixing drain water with the same volume of total ionic strength adjustment buffer solution. The stress–strain characteristics were measured using a universal test machine (3340, Instron Corporation, Norwood, MA, USA) with a strain rate of 5 mm·min^−1^ for a specimen of 2 × 1 cm (length by width) at room temperature.

### 2.4. Fuel Cell Performance Measurements and Electrochemical Analysis

The prepared MEA was placed in the center of a single cell and assembled with other components such as GDLs (39BB, SGL Carbon, Wiesbaden, Germany), Teflon gaskets (250 µm), bipolar plates with serpentine channels of 1 mm in width and height, and end plates. The single-cell assembly was tightly assembled by tightening eight bolts with 80 in·lb torque. For fuel cell performance measurement, a single cell was connected to a PEMFC station (CNL, Seoul, Korea). The activation process was performed for 2 h with repeating constant voltage steps of 0.8, 0.6, and 0.3 V for 3 min at 70 °C and 100% relative humidity (RH) by supplying hydrogen (H_2_) (150 mL·min^−1^) and air (800 mL·min^−1^) to the anode and cathode, respectively, without back pressure. Then, the polarization curve was obtained by the current sweep method with a rate of 50 mA·s^−1^ from open-circuit voltage (OCV) to 0.3 V under the same environmental conditions of the activation process. The electrochemical impedance spectroscopy (EIS) spectra were obtained at a frequency range of 0.1 Hz–15 kHz and an amplitude of 10 mV at 0.6 V, and the experimental conditions were the same as those in the polarization tests with an impedance analyzer (HCP-803, BioLogic, Seyssinet-Pariset, France). For the calculation of the electrochemical active surface area (ECSA), the cyclic voltammetry (CV) spectra were obtained by a potentiostat (HCP-803, BioLogic, France) under a potential range of 0.1–1.2 V at a scanning speed of 100 mV·s^−1^ by supplying hydrogen (H_2_) (50 mL·min^−1^) and nitrogen (N_2_) (200 mL·min^−1^) to the reference (anode) and working (cathode) electrodes. After changing the hydrogen flow rate to 200 mL·min^−1^, linear sweep voltammetry (LSV) was conducted to determine the hydrogen crossover current density at 0.1–0.6 V with a scanning rate of 2 mV·s^−1^.

An OCV holding test was conducted for 72 h while supplying 300 mL·min^−1^/300 mL·min^−1^ of hydrogen (H_2_) and oxygen (O_2_) at 30% RH and 90 °C, respectively. After 72 h OCV holding test, the drain water from the anode and cathode was collected to analyze the concentration of total eluted fluorine ions.

### 2.5. Characterizations

The change in elemental compositions of the MWCNTs before and after the acid treatment was measured using an X-ray photoelectron spectrometer (Sigma Probe, Thermo Fisher Scientific) with Al Kα as the X-ray source. The morphological properties of CeO_2_ were investigated using a transmission electron microscope (TEM) (JEM-ARM200F, JEOL, Tokyo, Japan) with an acceleration voltage of 200 kV. The surface and cross-sectional scanning electron microscope (SEM) images were obtained using a field-emission SEM (SU-5000, HITACHI, Tokyo, Japan) with an accelerating voltage of 5 kV. Thermogravimetric analysis (TGA) was conducted at a temperature range of 30–900 °C and a heating rate of 10 °C·min^−1^ by supplying an air/N_2_ flow rate of 60/40 mL·min^−1^ for the MWCNTs/CeO_2_ and an N_2_ flow rate of 100 mL·min^−1^ for the prepared membranes. The proton conductivity was measured by cutting the membranes to a size of 1 × 4 cm and inserting them into the conductivity cell, where four Pt wire was located at a distance of 1 cm for each electrode. The membrane resistance was obtained by EIS measurement in the frequency range from 1 MHz to 1 Hz at a constant current of 0.1 mA at a temperature of 30 °C and 70 °C by supplying fully humidified nitrogen gas. The in-plane conductivities of membranes were calculated using the measured resistance value and the following equation [22]:(4)σ=L/RA
where *L*, *R*, and *A* represent the distance between the electrodes, membrane resistance, and cross-sectional area of the membrane, respectively. The dimensional stability and water uptake capacity of the prepared membranes were calculated as follows.
(5)$$\mathrm{Length}\;\mathrm{change}\;\left(\%\right)=100\left(\frac{L_{f}-L_{i}}{L_{i}}\right)$$
(6)$$\mathrm{Thickness}\;\mathrm{change}\;\left(\%\right)=100\left(\frac{t_{f}-t_{i}}{t_{i}}\right)$$
(7)$$\mathrm{Water}\;\mathrm{uptake}\;\left(\%\right)=100\left(\frac{m_{f}-m_{i}}{m_{i}}\right)$$

The membrane specimens were cut to a size of 1 × 1 cm and dried in a vacuum oven at 80 °C for 12 h. After cooling down to an ambient temperature of 25 °C, the membrane specimens were taken out from the oven, and initial mass (m*_i_*), length (L*_i_*), and thickness (t*_i_*) were measured. After that, the samples were submerged in DI water at 80 °C for 12 h in a convection oven. The final mass (m*_f_*), length (L*_f_*), and thickness (t*_f_*) of the membranes were measured by removing the residue water onto the surface of the membranes.

## 3. Results and Discussion

Figure 1a–e illustrate the fabrication processes of the MEA with the modified membrane with the ultrathin MWCNTs/CeO_2_ functional layers. First, to prepare the uniformly dispersed mixed solution of MWCNTs/CeO_2_ for spray coating, a chemical acid treatment process for modifying the MWCNTs was conducted (Figure 1a). Compared to the pristine MWCNTs, the MWCNTs with acid treatment exhibited significantly higher O1s peaks at 530.8 and 532.5 eV corresponding to the C=O and C–O–H functional groups, respectively, as shown in Appendix A [40,41,42,43]. The introduction of the oxygen functional groups can change the wetting properties of the hydrophobic MWCNTs to hydrophilic, which improves the dispersion characteristics of the MWCNTs. To visually compare the dispersibility, the MWCNTs with and without acid treatment were added to each vial containing an IPA solution and treated with ultrasonication for 30 min (Appendix A). Immediately after the ultrasonication, the undispersed and agglomerated pristine MWCNTs stuck to the vial surface. On the other hand, the MWCNTs with acid treatment showed superior dispersion properties without any noticeable agglomeration problem. After chemical modification of the MWCNTs used for improving the mechanical properties of the membrane, successive processes for preparing the mixed solution of MWCNTs/CeO_2_ were conducted through the mechanical ball-milling process (Figure 1b) to secure the chemical durability of the membrane. Generally, CeO_2_ has been considered the most promising HO^•^ radical scavenger because only 1% of the cerium ions in the sulfonic acid groups in the membrane exhibit a significant quenching ability of about 89% [38]. However, the agglomerated nanosized CeO_2_ in the membrane can severely reduce proton transport. With the aid of the high-energy ball-milling process [44,45,46], the hydrophilic CeO_2_ nanoparticles mechanically collided repeatedly with the MWCNTs with the oxygen functional groups. During this process, the CeO_2_ nanoparticles could adhere to the modified MWCNTs surfaces and, therefore, could be located on 3D MWCNTs fiber networks, which made the solution have uniformly dispersed MWCNTs/CeO_2_ without agglomeration. However, without the ball-milling process, agglomeration of the CeO_2_ nanoparticles that floated on the solution was observed, as shown in Appendix A. To secure the proton transport in the void space of a 3D fiber network of MWCNTs/CeO_2_, the Nafion ionomer was added to the solution and a very short ultrasonication (under 1 min) was conducted to improve the ionomer distribution while minimizing the detachment of CeO_2_ from the MWCNTs (Figure 1c). After completely preparing the solutions, the ultrathin MWCNTs/CeO_2_ layers were constructed. Figure 1d shows the deposition of the ultrathin MWCNTs/CeO_2_ functional layers on the Nafion membrane outer surface through an automatic lab-made system consisting of a vacuum hotplate and a 3D-printer-based spray coater. By a layer-by-layer coating of the diluted solution, the MWCNTs/CeO_2_ could be stacked by forming a 3D fiber network with uniform distribution, and the void spaces were filled with the Nafion ionomer, which could effectively secure the proton transport at the membrane/electrode interface. To improve the uniformity of the functional layers, the spray travel path was designed to have four serpentine-like patterns and the center position was repeatedly shifted. Furthermore, to precisely control the coating thickness, the number of coating and deposition times was carefully controlled. Figure 1e depicts the introduction of the modified Nafion membrane with the ultrathin MWCNTs/CeO_2_ functional layers, and Figure 1f shows the completed MEA with the modified membrane. Because the functional layers were only incorporated onto the membrane surface, it was expected that the layers would efficiently remove the radicals at the membrane/electrode interface where radical attack severely occurs and have an effective role in improving the mechanical properties while minimizing the use of non-ion-conducting fillers.

Figure 2a,b show the SEM surface images of the MWCNTs/CeO_2_ with a weight ratio of MWCNTs:CeO_2_ of 4:1. Because the Nafion ionomer filled the void space and made it difficult to properly observe the MWCNTs and CeO_2_, here, the MWCNTs/CeO_2_ solution without the Nafion ionomer was deposited after the ball-milling process. The magnified SEM image in Figure 2b shows that CeO_2_ is uniformly dispersed on a well-distributed 3D MWCNTs fiber network without severe agglomeration. To find out the proper weight ratio between the MWCNTs and CeO_2_, another sample of MWCNTs/CeO_2_ with a weight ratio of MWCNTs:CeO_2_ of 4:1 was prepared for comparison. Appendix A shows the surface backscattered electron images of the functional layers with weight ratios of MWCNTs:CeO_2_ of 2:1 (Appendix A) and 4:1 (Appendix A). In the case of MWCNTs:CeO_2_ = 2:1, in which the weight ratio of CeO_2_ to the MWCNTs was two times larger compared to MWCNTs:CeO_2_ = 4:1, aggregated CeO_2_ particles were observed despite the high-energy ball-milling process due to the weak repulsive force and the intrinsic instability of the nanoparticles [38,47]. Given that CeO_2_ aggregation reduces the radical scavenging efficacy and proton conductivity, MWCNTs:CeO_2_ = 4:1 with a highly dispersed CeO_2_ distribution was selected. From the TEM measurement (Figure 2c) and the corresponding energy-dispersive X-ray analysis (Appendix A), the morphological features of the commercial CeO_2_ nanoparticles with a size of ∼25 nm and the elemental composition were confirmed. Figure 2d shows the TGA of the synthesized MWCNTs/CeO_2_ with a weight ratio of 4:1. Although 20 wt% of CeO_2_ relative to the MWCNTs was added, at a temperature of 800 °C, only 14 wt% of CeO_2_ remained, which is less than 20 wt%. This is considered to be due to the thermal decomposition of the CeO_2_ nanoparticles, absorbed solvents, and organic substances from the commercial CeO_2_ dispersed solutions [48,49,50]. Additionally, with the rapid increase in the thermal decomposition of the MWCNTs at above 600 °C, it can be seen that the change in bulk properties of the MWCNTs is not significant from the introduction of defects or functional groups onto the surface by the acid treatment process [51]. In addition, the comparison of the thermal behavior of pristine NR211 and modified NR211 with MWCNTs/CeO_2_ layers was further conducted as shown in Appendix A. The TGA curves under N_2_ atmosphere show the decomposition behavior of the membranes with a weight loss of three stages. The first weight loss was observed at approximately 150 °C, which was related to the evaporation of absorbed bound water. The second one was ascribed to the decomposition of the sulfonic acid group at approximately 350 °C. Subsequently, the decomposition of the polymer main chain begins at approximately 450 °C. The weight loss of NR211 with MWCNTs/CeO_2_ membrane was much lower than that of NR211 below 350 °C because increased interfacial interactions between the MWCNTs/CeO_2_ layers and NR211 led to an increase in the thermal degradation activation energy [52,53]. Therefore, the thermal stability of NR211 with MWCNTs/CeO_2_ layers exhibited higher thermal stability up to 400 °C compared to the reference NR211. Figure 2e,f show the cross-sectional SEM images of the MWCNTs/CeO_2_ with different amounts of spraying solution (15 and 30 mL). As the amount was doubled, the thickness also linearly increased from 450 to 830 nm. This indicates that precise control of the coating thickness in the submicrometer scale is possible with the use of the layer-by-layer spray-coating process with diluted MWCNTs/CeO_2_ solutions. To confirm the proper coating thickness of the MWCNTs/CeO_2_ layer, electrochemical measurements were conducted by incorporating them into each MEA, and a severe performance reduction in the relatively thick MWCNTs/CeO_2_ layer of 830 nm was observed (Appendix A and Appendix A). On the other hand, in the case of the MWCNTs/CeO_2_ layer with a thickness of 450 nm, because it showed an initial performance drop of less than 8.1% compared to the reference MEA without MWCNTs/CeO_2_ layers, we chose 450 nm as the proper thickness for mechanically and chemically reinforcing the membrane without severe performance reduction.

Since the introduction of a functional layer can affect the ion conductivity and the mechanical properties of the membrane, membrane characterization such as in-plane proton conductivity, dimensional stability (length and thickness change), water uptake, and stress–strain behavior were investigated. As shown in Table 1, the modified NR211 with ultrathin MWCNTs/CeO_2_ layers showed a marginally reduced ion conductivity by ∼2.6% at 30 °C and ∼3.4% at 70 °C compared to the reference NR211. In previous studies where the CNTs [23] and CeO_2_ [34,37] were simply mixed with electrolyte ionomer during membrane fabrication, it was confirmed that the ion conductivity was severely reduced by over ∼15 %. In Kim’s recent study [37], the CRE/Ce structure effectively alleviated cerium ion migration during PEMFC operation; however, the ion conductivity of the modified membrane was severely reduced by ∼35%. Therefore, the selective introduction of the MWCNTs/CeO_2_ layers at the outermost surfaces of the membrane with an ultrathin thickness can effectively alleviate the conductivity loss of the membrane when incorporating nonconductive reinforcing materials. When it comes to dimensional stability, the modified NR211 with ultrathin MWCNTs/CeO_2_ layers swelled more in a thickness direction (*z* axes) than the reference NR211. On the other hand, the dimensional changes in the length direction (*x* and *y* axes) were decreased. These dimensional changes indicate that the MWCNTs/CeO_2_ layer affected the structural stability of the membrane in a wet state. Furthermore, the modified membrane exhibited increased water uptake capacity due to the hydrophilic properties of CeO_2_ and acid-treated MWCNTs, which let the membrane retain water molecules on the surface.

Figure 3a,b show the comparison of the stress–strain curves and the FERs after Fenton’s tests for the reference NR211 and the modified NR211 with the ultrathin MWCNTs/CeO_2_ layers, respectively. To evaluate the improved mechanical properties and chemical durability of the modified membrane, the stress–strain curves were obtained, and Fenton’s tests were conducted. The Young’s modulus of the modified NR211 with the ultrathin MWCNTs/CeO_2_ layers (408 MPa) is much higher than that of the reference NR211 (340 MPa) as shown in the stress–strain curves in Figure 3a. Furthermore, shortened elongation at break signifies an increase in hardness of the modified membrane. These improved mechanical properties are due to the mechanical support of the MWCNTs at both sides of the membrane, and they can provide the membrane higher resistance to dimensional change during repeated wet/dry cycles of the PEMFCs. Next, an ex situ Fenton’s test was conducted to evaluate the radical scavenging effect of the MWCNTs/CeO_2_ at the Nafion membrane surfaces (Figure 3b). The prepared specimens were immersed in the Fenton solution (30 ppm Fe^2+^ in 20 wt% aqueous H_2_O_2_) where free radicals are generated continuously by the following equations [54,55]:(8)$${\mathrm{Fe}}^{2+}+H^{+}+H_{2}O_{2}\to {\mathrm{Fe}}^{3+}+H_{2}O+{\mathrm{HO}}^{•}$$
(9)$${\mathrm{Fe}}^{3+}+H_{2}O_{2}\to {\mathrm{Fe}}^{2+}+H^{+}+{\mathrm{HOO}}^{•}.$$

The generated free radicals in the H_2_O_2_ solution chemically decompose the main and side chains of the Nafion membrane [34]. By measuring the fluoride ion concentration derived from the decomposition of the membrane, the FER was calculated by the following equation [56,57,58]:(10)$$\mathrm{FER}=\frac{C\times V}{t\times m\times M}\;\left(µ\mathrm{mol}\cdot h^{-1}\cdot g^{-1}\right)$$
where C is the fluoride ion concentration (g·cm^−3^), V is the volume of the solution (cm^−3^), t is the time of the test (h), m is the weight of the dried membrane before the test (g), and M is the molecular weight of fluorine (g·mol^−1^). In Figure 3b, the modified Nafion membrane with MWCNTs/CeO_2_ shows 72.56% lower FER (3.26 µmol·h^−1^·g^−1^) than the reference NR211 (11.88 µmol·h^−1^·g^−1^), which means that the MWCNTs/CeO_2_ layer effectively mitigates the chemical degradation of the membrane by scavenging radicals at the membrane’s outermost region. Figure 3c,d show the magnified SEM surface images of the reference and modified membranes after the Fenton’s test, respectively. Interestingly, the reference NR211 membrane specimen was ripped or formed large pinholes as a result of the chemical degradation; however, the degradation of the modified membrane was severely restricted because of the existence of CeO_2_ in the functional layers. The key values for the measurements are summarized in Table 2.

To evaluate the performance and durability of the MEAs with the prepared membranes, single cells were constructed with only differences in the membranes, and the electrochemical characteristics were measured. Figure 4a depicts the OCV decay spectra over 72 h under accelerated conditions to compare the membrane’s chemical degradation rates due to the free radicals generated during the OCV holding test [59,60]. The OCV holding protocol was used to investigate the difference in chemical durability between the reference NR211 and the modified NR211 with the MWCNTs/CeO_2_ functional layers. The OCV holding test was conducted at a high temperature of 90 °C and a low RH of 30% by supplying 300 sccm of H_2_ and O_2_ to the anode and cathode, respectively. The MEA with the modified membrane with the MWCNTs/CeO_2_ functional layers exhibited a much lower decay rate of 0.931 mV·h^−1^ than the MEA with a pristine NR211 membrane (1.861 mV·h^−1^). This indicates that the very low loading of CeO_2_ in the ultrathin functional layers can exhibit the role of an effective radical scavenger at both sides of the membrane (membrane/electrode interface). In addition, both samples started at 1.010 V in a well-polarized state, which confirmed that the incorporation of the MWCNTs/CeO_2_ functional layers did not damage or change the bulk characteristics of the Nafion membrane and was an effective strategy for engineering the interface between the membrane and electrode of the MEA. Figure 4b,c show the polarization curves before and after the OCV holding test at 70 °C and 100% RH with a H_2_/air flow rate of 150/800 mL·min^−1^ and no backpressure, respectively. In the case of the initial MEA performances, the MEA with the modified membrane shows 7.28% inferior peak power density (PPD) compared to the MEA with the reference NR211 because the addition of the functional layers inevitably increased the ohmic resistance and charge transport resistance at the interface, as shown in Figure 4d and Table 3. However, the increases in ohmic resistance (3.93%) and charge transport resistance (2.22%) were sufficiently restricted because of the ultrathin layer thickness (450 nm), the well-dispersed CeO_2_ on the 3D MWCNTs fiber network, and filling the ionomer in the void space. Notably, following 72 h of the OCV holding test, the MEA with the modified membrane exhibited higher PPD and current density at 0.6 V by 17.9% and 36.9%, respectively, compared to those of the MEA with the reference NR211 due to the improved chemical durability from the MWCNTs/CeO_2_ functional layers. The corresponding EIS spectra confirmed that the increased rates in ohmic resistance and charge transport resistance for the MEA with the modified membrane were not large compared to those of the reference after the durability test (Figure 4e and Table 3). In particular, when compared to the MEA with the modified membrane, the significant increase in charge transport resistance of the reference MEA during the OCV holding test further confirmed that the radical-attack-induced chemical degradation of the membrane severely deteriorated the charge transport at the membrane/electrode interface.

Figure 5a,b show the LSV spectra before and after the OCV holding test to compare the hydrogen crossover current densities of the MEAs, respectively. In the case of the initial hydrogen crossover current density, the MEA with the MWCNTs/CeO_2_ functional layers had a 22% lower value (1.75 mA·cm^−2^) compared to the reference MEA (2.25 mA·cm^−2^). This signifies that the addition of the MWCNTs/CeO_2_ functional layers on both sides of the membrane acted as an effective gas barrier (Figure 5a). Notably, after the OCV holding test, the difference in hydrogen crossover current density further increased to 33% because of the improved chemical durability of the modified membrane. Figure 5c,d show the CV spectra before and after the OCV holding test, respectively; almost the same ECSA values were calculated for the MEAs with the reference and modified membranes because the same catalyst and catalyst loading amounts were used in the experiments. Moreover, the differences in ECSA before and after the OCV holding test were insignificant, which means that the electrode suffered little damage compared to the highly damaged membrane during the durability test. The summarized values are shown in Table 4.

Figure 6a,b show the cross-sectional SEM images of the MEAs with the reference NR211 and the modified NR211 with the MWCNTs/CeO_2_ functional layers after the OCV holding test, respectively. The thickness of the NR211 membrane was reduced by over 7 µm in the case of the reference MEA, whereas there was no difference from the original thickness for the modified NR211 with the MWCNTs/CeO_2_ during the OCV holding test. These results further confirmed the improved chemical durability of the modified membrane due to the existence of CeO_2_ in the functional layer and supported the results of both the OCV decay rate spectra and the variation in *I*–*V* performance and hydrogen crossover current density after the OCV holding test. In addition, there were almost no differences in the thicknesses of the electrodes in both MEAs, which further confirmed the similar ECSA values in the CV spectra after the OCV holding test. To qualitatively compare the membrane degradations using the emission of fluoride ions from the decomposition of the membrane from the radical attack during the OCV holding test, concentrations of fluoride ions in the drain water at the anode and cathode were measured (Figure 6c). In the case of the reference MEA, accumulated concentrations of fluoride ions for 72 h were calculated as 6.27 ppm (anode) and 6.85 ppm (cathode), and these values were much higher than those of the MEA with the modified membrane (0.261 and 0.163 ppm for the anode and cathode, respectively). This means that the reference membrane was severely damaged by the radical attack during the OCV holding test, which further confirmed the results of severe membrane thickness reduction for the reference membrane, as well as the efficacy of introducing the ultrathin functional layers for reinforcing the membrane.

## 4. Conclusions

In this study, a strategy for introducing ultrathin MWCNTs/CeO_2_ functional layers at both sides of a Nafion membrane was developed for minimizing conductivity loss while improving the mechanical and chemical durability of the membrane. Through the chemical modification of the MWCNTs with acid treatment, a high-energy ball-milling process of mixed solutions of MWCNTs and CeO_2_, and optimized spray coating conditions, a well-distributed 3D MWCNTs/CeO_2_ network with a thickness of ∼450 nm was successfully constructed on the commercial NR211 membrane. The modified NR211 with ultrathin MWCNTs/CeO_2_ layers exhibited a marginal proton conductivity loss of less than 3.4% despite the addition of nonconducting reinforcing agents while improving dimensional stability, mechanical properties, and chemical stability compared to those of pristine NR211. Furthermore, the ultrathin layer exhibited reduced hydrogen crossover by acting as an effective gas barrier at the outermost surfaces of the membrane. The MEA with the modified membrane had a slight decrease in initial performance but showed significantly improved chemical durability by exhibiting much slower OCV decay rates and a significant reduction in fluoride ion emission rates during an accelerated 72 h OCV holding test. We believe that this simple methodological approach to modify commercial membranes to improve mechanical and chemical durability can contribute to advances in commercializing PEMFCs by enabling long-term operation.

## Figures and Tables

**Figure 1 membranes-12-01073-f001:**
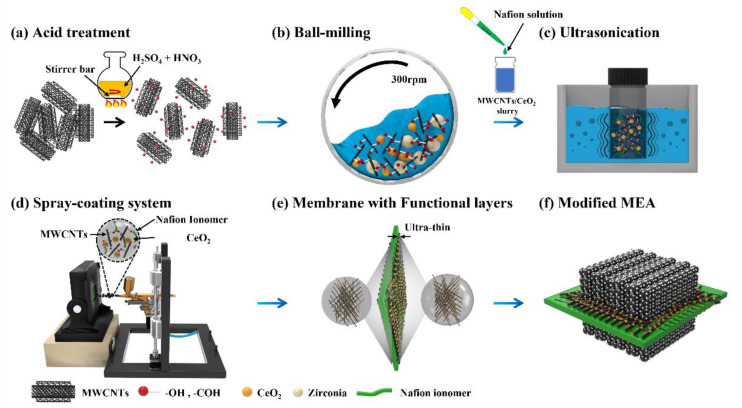
Schematic diagrams of preparing the MEA with the ultrathin MWCNTs/CeO_2_ functional layers. (**a**) Acid treatment process of the MWCNTs. Dispersion processes of MWCNTs/CeO_2_ through (**b**) successive mechanical ball-milling process and (**c**) ultrasonication. (**d**) Spray-coating process. (**e**) Mechanically and chemically reinforced Nafion membrane with the ultrathin MWCNTs/CeO_2_ layers at both sides. (**f**) Completed MEAs with the modified membrane.

**Figure 2 membranes-12-01073-f002:**
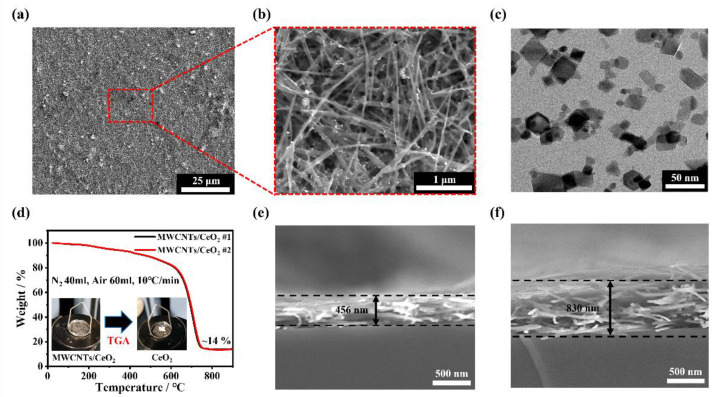
SEM surface images of the MWCNTs/CeO_2_ without the Nafion ionomer with magnification at (**a**) 1 k and (**b**) 30 k. (**c**) TEM image of the CeO_2_ nanoparticles. (**d**) TGA results of the MWCNTs/CeO_2_. Cross-sectional SEM images of the MWCNTs/CeO_2_ with thicknesses of (**e**) 450 nm and (**f**) 830 nm.

**Figure 3 membranes-12-01073-f003:**
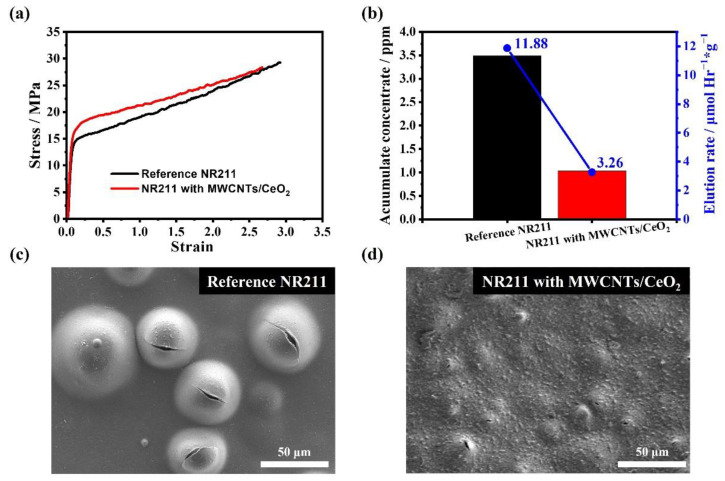
Comparison of (**a**) the stress–strain curves and (**b**) the fluorine emission rates (FERs) after Fenton’s test for the reference NR211 and the modified NR211 with MWCNTs/CeO_2_. Magnified SEM surface images of (**c**) the reference NR211 and (**d**) the modified NR211 with MWCNTs/CeO_2_ after Fenton’s test.

**Figure 4 membranes-12-01073-f004:**
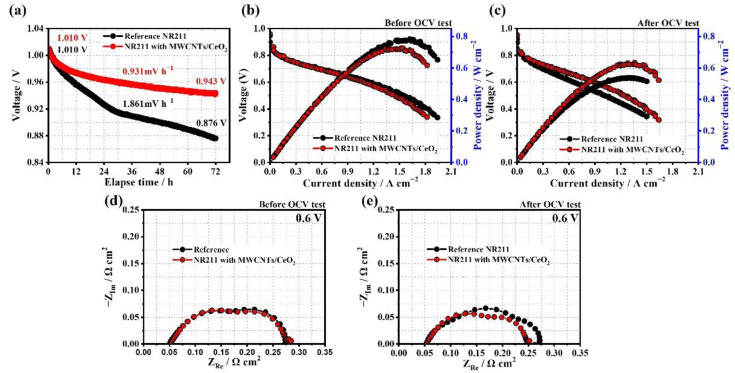
(**a**) OCV decay spectra over 72 h under accelerated conditions. Polarization curves (**b**) before and (**c**) after the OCV holding test. Corresponding EIS spectra at 0.6 V (**d**) before and (**e**) after the OCV holding test.

**Figure 5 membranes-12-01073-f005:**
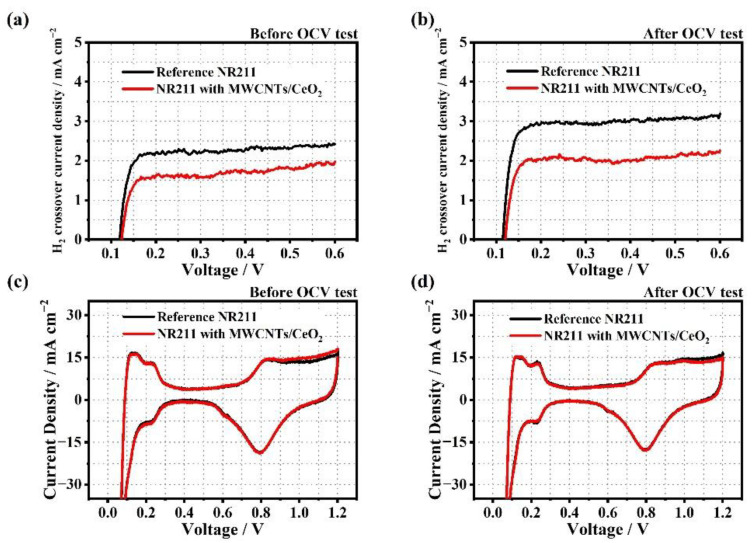
LSV spectra of the MEAs with the reference NR211 and the modified NR211 (**a**) before and (**b**) after the OCV holding test. CV spectra of the MEAs with the reference NR211 and the modified NR211 (**c**) before and (**d**) after the OCV holding test.

**Figure 6 membranes-12-01073-f006:**
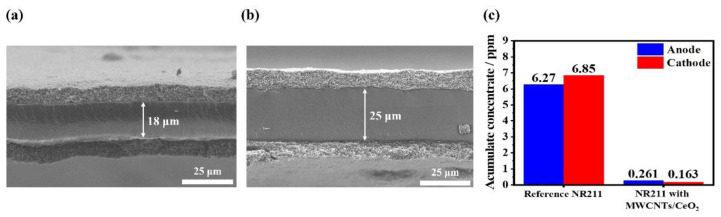
Cross-sectional SEM images of the MEAs with (**a**) the reference NR211 and (**b**) the modified membrane with the MWCNTs/CeO_2_ functional layers after the OCV holding test. (**c**) Accumulated concentration of fluoride ions in the drain water after the OCV holding test.

**Table 1 membranes-12-01073-t001:** Summary of ion conductivity, dimensional stability, and water uptake.

Samples	Proton Conductivity		Dimensional Stability (%)			Water Uptake (%)
	30 °C (S/cm)	70 °C (S/cm)	*X*-Axis	*Y*-Axis	*Z*-Axis	
Reference NR211	0.1299 (-)	0.2369 (-)	11.0	12.5	12.0	20.0
NR211 with MWCNTs/CeO_2_	0.1265 (−2.6%)	0.2289 (−3.4%)	8.7	11.2	17.9	22.4

**Table 2 membranes-12-01073-t002:** Summary of the stress–strain curves and Fenton’s test.

	Tensile Strength (MPa)	Young’s Modulus (MPa)	Elongation at Break (%)	Accumulate Concentrate (ppm)	Fluorine Emission Rate (μmol h^−1^ g^−1^)
Reference NR211	29.30	340.23	292	3.49	11.88
NR211 with MWCNTs/CeO_2_	28.34	408.29	268	1.03	3.26

**Table 3 membranes-12-01073-t003:** Summary of the key parameter values.

Sample	OCV (V)	Current Density (A cm^−2^)	Peak Power Density (W cm^−2^)	R_ohm_ (Ω cm^2^)	R_ct_ (Ω cm^2^)
**Before OCV test**					
Reference NR211	0.952	1.16 (-)	0.783 (-)	0.0508 (-)	0.2248 (-)
NR211 with MWCNTs/CeO_2_	0.960	1.08 (−6.90%)	0.726 (−7.28%)	0.0528 (+3.93%)	0.2299 (+2.22%)
**After OCV test**					
Reference NR211	0.922	0.65 (-)	0.537 (-)	0.0531 (-)	0.2174 (-)
NR211 with MWCNTs/CeO_2_	0.953	0.89 (+36.92%)	0.633 (+17.87%)	0.0535 (+0.75%)	0.1971 (−9.34%)

**Table 4 membranes-12-01073-t004:** Summary of the ECSA and hydrogen crossover current at 0.4 V before and after the OCV holding test.

Sample	Before OCV Test		After OCV Test	
	LSV (mA cm^−2^)	ECSA (m^2^ g_pt_^−1^)	LSV (mA cm^−2^)	ECSA (m^2^ g_pt_^−1^)
Reference NR211	2.25	46.95	3.00	41.44
NR211 with MWCNTs/CeO_2_	1.75	45.95	2.00	41.00

## Data Availability

The data presented in this study are available on request from the corresponding author.

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
