# Peer review of "Reinforced Nafion Membrane with Ultrathin MWCNTs/Ceria Layers for Durable Proton-Exchange Membrane Fuel Cells"

_membranes, 2022, doi:10.3390/membranes12111073_

Round 1

Reviewer 1 Report

The paper presents work on Reinforced Nafion Membrane with Ultrathin MWCNTs/Ceria Layers for Durable Proton-Exchange Membrane Fuel Cells, the paper looks interesting, however, the author should consider the following points:

- Improve the introduction and literature review, author should look at recent published papers in Energy Field, you need to look at recent published work on PEMFC.

-Methodology and experimental procedures should be described in details.

Discussions should show the contribution and compare with other results as mentioned above, for example what about other research on membrane developments.

Conclusions should show clearly the contribution of this work.

Reviewer 2 Report

The manuscript reported the strategy for modifying commercial Nafion membranes by introducing ultrathin MWCNTs/CeO2 layers on both sides of the membrane to construct a mechanically and chemically reinforced MEA. The characteristics of the novel incorporated layers and the reinforced Nafion membrane were investigated using various in-situ and ex-situ measurement techniques. Moreover, the electrochemical measurements for fuel cells were conducted.

I consider the content of this manuscript will definitely meet the reading interests of the readers of the Membranes journal. However, there are certain English spelling and grammar issues, and also the discussion and explanation should be further improved.

Therefore, I suggest giving a minor revision and the authors need to clarify some issues or supply some more experimental data to enrich the content

Detailed comments can be found in the PDF file.
